

# Reconciling the Apparent Absence of a Last Glacial Maximum Alpine Glacial Advance, Yukon Territory, Canada, through Cosmogenic Beryllium-10 and Carbon-14 Measurements

Brent M Goehring[1], Brian Menounos[2], Gerald Osborn[3], Adam Hawkins[2], Brent Ward[4]

[1]Department of Earth and Environmental Sciences, Tulane University, New Orleans, LA 70118 USA
[2]Geography Program and National Resources and Environmental Studies Institute, University of Northern British Columbia, Prince George, BC Canada
[3]University of Calgary, Calgary, AB Canada
[4]Simon Fraser University, Burnaby, BC Canada

*Correspondence to*: Brent Goehring (bgoehrin@tulane.edu)

**Abstract.** We present a new in situ produced cosmogenic beryllium-10 and carbon-14 nuclide chronology from two sets (outer and inner) of alpine glacier moraines from the Grey Hunter massif of southern Yukon Territory, Canada. The chronology potential of moraines deposited by alpine glaciers outside the limits of the Last Glacial Maximum (LGM) ice sheets potentially provide a less-ambiguous archive of mass balance, and hence climate than can be inferred from the extents of ice sheets themselves. Results for both nuclides are inconclusive for the outer moraines, with evidence for pre-LGM deposition (beryllium-10) and Holocene deposition (carbon-14). Beryllium-10 results from the inner moraine are suggestive of canonical LGM deposition, but with relatively high scatter. Conversely, in situ carbon-14 results from the inner moraines are tightly clustered and suggestive of terminal Younger Dryas deposition. We explore plausible scenarios leading to the observed differences between nuclides and find that the most parsimonious explanation for the outer moraines is that of pre-LGM deposition, but many of the sampled boulder surfaces were not exhumed from within the moraine until the Holocene. Our results thus imply that the inner and outer moraines sampled pre- and post-date the canonical LGM and that moraines dating to the LGM are lacking likely due to overriding by the subsequent Late Glacial/earliest Holocene advance.

## 1 Introduction

We present an in situ cosmogenic beryllium-10 and carbon-14 chronology of alpine glacier advances from interior Yukon territory that escaped inundation by the Cordilleran Ice Sheet to infer climate conditions during the late Pleistocene. This is important as there are relatively few records of climate from the interior Yukon prior to the Holocene (e.g., Mahony, 2015). Glaciers, both alpine and ice sheets, are commonly viewed as sensitive indicators of climate change, both past and present, as they integrate conditions affecting their mass balance and thus reflect regional climate (e.g., temperature and precipitation). Yet, the chronology of ice sheet margins may differ from those of alpine glaciers because of differences in response time to climate perturbations (Jóhannesson et al., 1989), differences in ice dynamics including changes in the position of ice divides



(Fulton, 1991), and effects of debris cover on alpine glaciers (Scherler et al., 2011). Additionally, the large size of an ice sheet means that its accumulation area may be far-removed from its margins and thus part of its mass balance is set by climate
different from that at the margin.

The margins are where we can date deposits left behind by former ice sheets. Studies exploring such climate reconstructions in the Yukon largely have focused on ice sheet margins, notably the northern margins of the Laurentide (LIS) and Cordilleran (CIS) ice sheets (e.g., Margold et al., 2013a; Stroeven et al., 2010; Stroeven et al., 2014) and found somewhat ambiguous results. Less studied are the alpine glaciers in the region occupying the many massifs and ranges that escaped inundation by
the LIS and CIS during glaciation, and that may provide a less ambiguous picture of climate forcing than records from ice sheet margins.

A common observation in cosmogenic nuclide derived chronologies of alpine glacier and ice sheet fluctuations is a general lack of coherence from a single feature, or multiple features of the same morphostratigraphic age (e.g., Balco, 2011; Heyman et al., 2011). Both old and young outliers are observed in the data sets, suggesting that pre- and post-depositional
processes are operating (Applegate et al., 2012; Applegate et al., 2010), such as inheritance of nuclides from earlier exposure periods or post-depositional shielding of the sampled surface (e.g., exhumation of the surface, ephemeral cover). The measurement of multiple nuclides with differing half-lives, ideally one long-lived and one short-lived, can potentially not only elucidate the age of the moraines but also provide insight into the processes generating the observed scatter in moraine surface exposure ages.

Our objective is to develop a chronology for moraines that head from the Grey Hunter massif in Yukon Territory, which escaped inundation by the Cordilleran Ice Sheet. Such a site allows for the reconstruction of climate fluctuations during the late Pleistocene derived from simpler, and likely more responsive, alpine glacier systems relative to the adjacent Cordilleran Ice Sheet. This paper describes new measurements of in situ cosmogenic beryllium-10 ($^{10}$Be) and carbon-14 ($^{14}$C) of moraine boulders from the Grey Hunter massif, Yukon Territory, Canada, delineating former alpine glacier extents.





**Figure 1. Map of the Grey Hunter Region delineating the Grey Hunter Massif as shown in Figure 2 and former CIS extents. Also shown is the Glenlyon Range.**

## 2.1 Study Area

### 1.2.1 Geology, Geomorphology, and Climate

Grey Hunter Peak (~2200 m) and its adjacent terrain, herein referred to as Grey Hunter massif, is located within the generally low-lying MacArthur Mountains in east-central Yukon. Grey Hunter massif is located adjacent to Tintina Trench and Tintina Fault, a large strike-slip fault that runs sub-parallel to the Denali Fault. The massif encompasses 450 km$^2$ of terrain that

preserves widespread evidence of Pleistocene glaciation as evidenced by the presence of cirques, aretes, and relatively wide, U-shaped valleys. We selected the Grey Hunter massif because previous work suggests it escaped inundation by the Cordilleran

ice sheet that flowed around the massif, but never overtopped it (Hughes, 1983). Today, the Grey Hunter massif hosts only small ice patches and rock glaciers; these features are limited to north-facing cirques. Grey Hunter massif consists predominantly of Cretaceous quartz monzonite of the Mayo Suite, which is the lithology of all collected samples.

Mean annual temperature and precipitation at the nearby Stewart Crossing weather station (63.62 °N, 135.87 °W, 504 m), is -4.3 °C and approximately 300 mm yr$^{-1}$. The mean elevation difference between Stewart Crossing and mean elevation at Grey

Hunter is approximately 1000 m. Assuming a simple moist adiabatic lapse rate of 6.5 °C km$^{-1}$, the mean annual temperature at Grey Hunter is -10.8 °C.

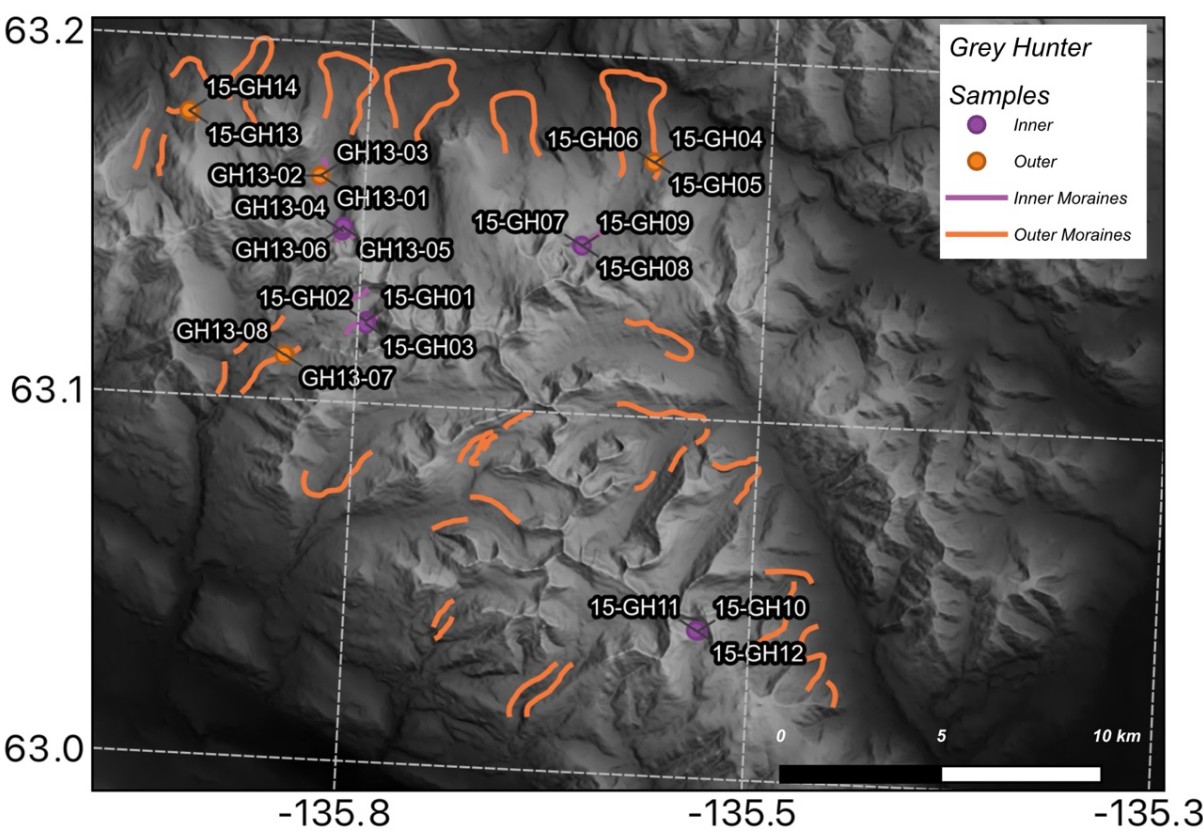

**Figure 2. Shaded relief map of the Grey Hunter Massif region. Outer moraines (orange) and inner moraines (purple) are shown. Sample locations are also shown for reference with the same color scheme.**

**4. Methods**

Samples were selected based on their prominence relative to the surrounding moraine surface. We chose large boulders (> 1 m$^3$) to minimize the chance of exhumation out of the moraine. Our sampling also targeted surfaces displaying signs of long-term exposure (e.g., lichen cover) over those with clean (e.g., unweathered, and non-lichen covered) surfaces that might be the



result of recent exhumation or boulder spalling. We collected samples using a gas-powered rock saw to extract rock 'brownies',

with approximate dimensions of 2 x 3 x 1 cm (L x W x H). A handheld GPS receiver was used to collect sample location and elevation, and we measured topographic shielding with a hand-held sighting clinometer by breaking the landscape into slope segments approximating the topography.

Sample preparation occurred in the Tulane University Cosmogenic Nuclide Laboratory. Samples were crushed, milled, and sieved to isolate the 250-500 micron size fraction. Quartz isolation followed standard laboratory procedures which includes

rinsing with tap water, magnetic separation, froth flotation to remove feldspars, and a minimum of two 24-hour 5%HF/5%HNO$_3$ etches on a shaker table and then two 24-hour 1%HF/1%HNO$_3$ etches in an ultrasonic bath (Nichols and Goehring, 2019). This method consistently produces quartz with Al and Ti concentrations less than 150 ppm and ensures complete removal of dodecylamine, which may interfere with in situ $^{14}$C analysis. Preparation of quartz separates for $^{10}$Be and in situ $^{14}$C analysis is the same for each nuclide. An aliquot of material for samples where $^{14}$C was also measured was split

from archived clean quartz material.

Chemical isolation of Be followed standard protocols, including sequential anion and cation columns (Ditchburn and Whitehead, 1994). All samples were spiked with ~0.25 mg Be (nominal $^{10}$Be/$^9$Be ratio ~4e-16). One process blank was prepared with each batch of eight samples. Be-isotope ratio measurements were made at the Purdue University Rate Isotope Measurement Laboratory (PRIME Lab), all normalized to dilutions as part of the standard KNSTD dilution series (Nishiizumi

et al., 2007).

We extracted carbon from quartz separates using a fully automated gas extraction system optimized for the fusion of quartz using LiBO$_2$ and collection, purification, and graphitization of evolved carbon as gaseous CO$_2$ (Goehring et al., 2019b). Process blanks are run every eighth sample. Additionally, the CRONUS-Earth CRONUS-A intercomparison material is run approximately every 40$^{th}$ sample to ensure tracking of long-term measurement stability and provide an estimate of sample

reproducibility, which typically exceeds individual sample analytical precision. Carbon isotope ratios were measured at the National Ocean Science Accelerator Mass Spectrometry (NOSAMS) facility at Woods Hole Oceanographic Institution relative to the primary standard Ox-II ($^{14}$C/$^{13}$C = 1.4575e-10, NIST SRM4990C). Primary and secondary (IAEA C7, $^{14}$C/$^{13}$C = 5.45e-11) standards are graphitized using the same reactors used for unknown samples, ensuring fully internal standardization of results.

Exposure ages are calculated using the scaling method for neutrons and protons outlined in Lifton et al. (2014), and a simplified muon scheme presented in Balco (2017), all of which are modulated by the geomagnetic model of Lifton (2016). Reference $^{10}$Be production rates were determined using the CRONUS-Earth "primary" calibration dataset (Borchers et al., 2016). For $^{14}$C production rates, we used the same production rate scaling method used for $^{10}$Be and calculated production rates by assuming that the CRONUS-A interlaboratory comparison standard (Jull et al., 2015; Goehring et al., 2019a) displays production-decay

saturation. All reported uncertainties below are at the 1-sigma level and include full propagation of analytical errors (quartz mass, carbon mass, AMS uncertainty, blank uncertainty).



## 5. Results

Two groups of moraines down valley of rock glaciers fringe many of the cirques of the study area (Figure 2). We refer to these two groups as the inner and outer moraines. Inner moraines are relatively sharp crested with a landform assemblage

characteristic of stability and antiquity greater than the LIA. Most boulders are prominent relative to their surroundings (Figure 3). No pronounced surface weathering of boulders is observed; most boulders have extensive, sometimes large lichen cover. Inner moraines typically extend 1.5 to 3 km down valley from cirque headwalls with a toe-headwall area ratio (THAR) equilibrium line altitude (ELA) estimate of 1565 m asl.

The outer moraines are decidedly more rounded in profile and boulders are generally more embedded, suggesting that boulder

exhumation occurred. Boulder weathering characteristics are more evolved than the inner moraine. Moraines extend 5.5 to 7 km down valley from cirque headwalls with a THAR derived ELA estimate of 1285 m asl.

Below we present resulting $^{10}$Be and $^{14}$C exposure ages, first from the outer moraines and then from the inner moraines (Tables S1, S2, and S3). Individual moraine medians, as well as group medians (outer and inner moraines) are presented in Table S4 with uncertainties represented by the half-width of the interquartile range. Where multiple $^{14}$C ages are finite (i.e., are not

saturated with respect to $^{14}$C production-decay systematics), we present the median age as above.

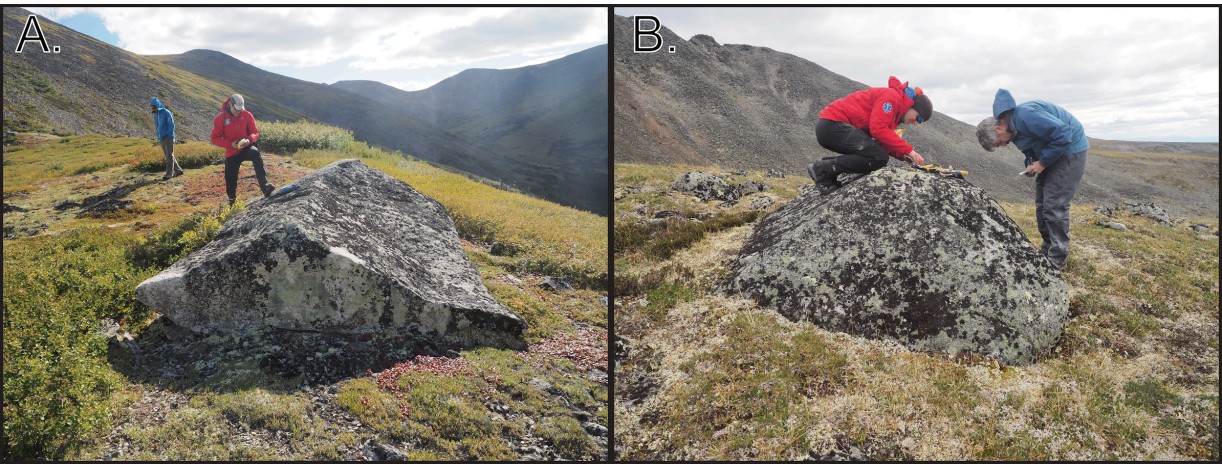

**Figure 3. Example of boulders sampled from outer moraines (A.) and inner moraines (B.). Note in A. the embedded character of the far-side of the large erratic.**

### 5.1 Outer Moraines

The four outer moraines yield median $^{10}$Be ages of 24.7 ± 0.6, 38.0 ± 7.7, 14.1 ± 8.5, and 38.2 ± 2.1 ka (Table S2, S3). For a given outer moraine, we generally observe a large degree of scatter (Figure 4); the one exception being sample set 1, with a resulting median age of 24.7 ± 0.6 ka. The quasi-standard error (ratio of half-interquartile range and median) for the outer moraine ranges from 2.5% to 60.3%. The grand group median for the outer moraines is 31.3 ± 8 ka.





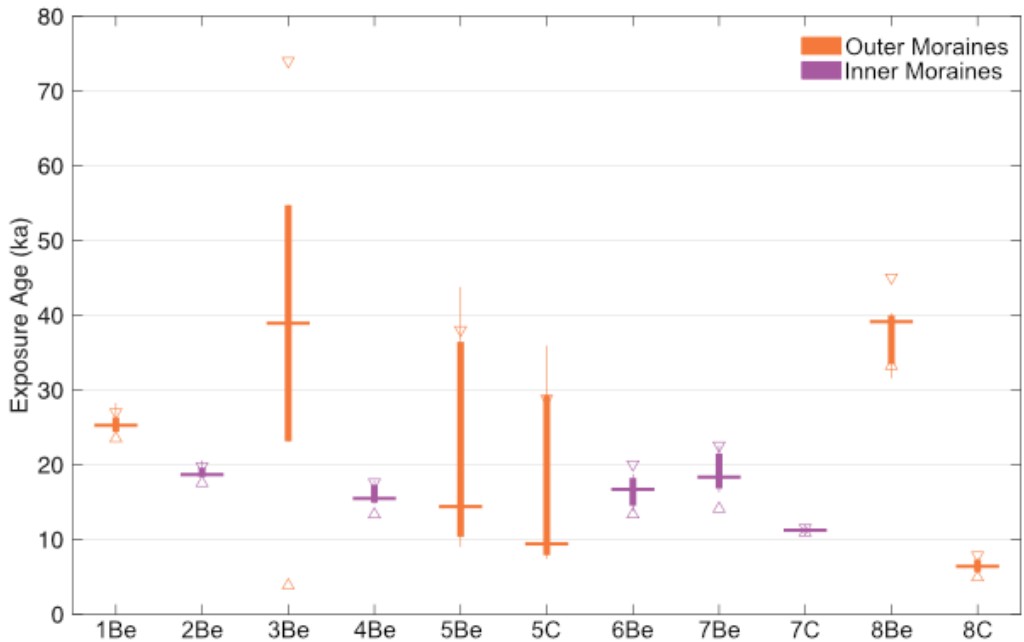

**Figure 4. Box and whisker plot of moraine ages for the inner moraines (purple) and the outer moraines (orange) as identified by their site number and nuclide. Note the larger magnitude of scatter and outlier spread for the outer moraines relative to the inner moraines. The larger spread for the carbon measurements from set 5 results from the measurement of one saturated age.**

In situ $^{14}$C results from sample set 5 yields one saturated (i.e., > ~30 kyr) exposure age, while two other samples yield finite ages of 9.4 ± 0.2 and 7.4 ± 0.2 ka. Similarly, all samples from sample set 8 yield finite $^{14}$C exposure ages of 7.6 ± 0.3, 5.4 ± 0.2, and 6.4 ± 0.2 ka. Results with respect to theoretical saturation concentrations for the elevation of a given sample is shown in Figure 5 along with isochrons of exposure duration. Determination of saturation for a sample is based on a statistically significant difference at one standard deviation between the theoretical saturation concentration and the measured sample concentration.





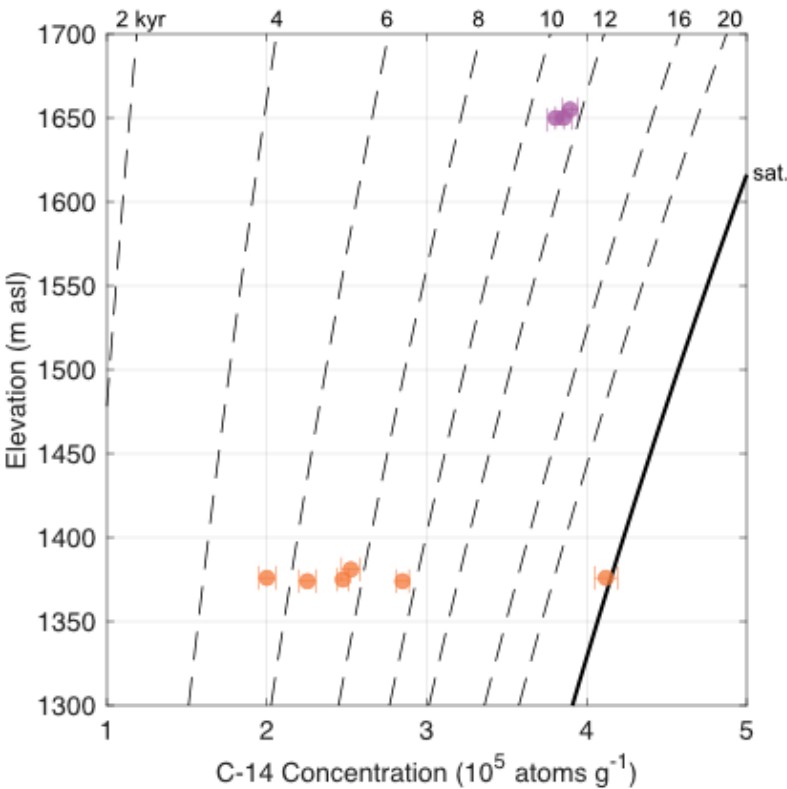

**Figure 5. Elevation vs $^{14}$C concentration. Contours show the equivalent concentrations for a given exposure age as a function of elevation. The heavy black line represents saturation concentration for $^{14}$C. The measurements from the inner moraine (purple) are tightly clustered relative to those of the outer moraines (orange).**

### 5.2 Inner Moraines

Resulting median $^{10}$Be exposure ages for the inner moraine sample sets are 18.2 ± 0.4, 15.1 ± 0.8, 16.3 ± 1.2, and 17.9 ± 1.5 ka (Figure 4; Table S4); the group median age is 17.1 ± 1.0 ka. The quasi-standard error as defined above ranges from 2.2% to 8.5%. Coherence for the younger moraines is better than that of the outer moraine sample sets. All measured samples from the younger moraine sample set 7 yields finite $^{14}$C exposure ages (Figures 4 and 5) with a median $^{14}$C age of 11.4 ± 0.1 ka (n = 3), and a quasi-standard error of 0.94%.

### 5.3 $^{14}$C-$^{10}$Be Isotope Ratios

A normalized paired-nuclide plot of $^{14}$C and $^{10}$Be concentrations can help identify complex exposure-burial histories (Figure 6). Concentrations are normalized by sample specific total nuclide production rates (spallation + muons) to yield concentrations equivalent to production rates of 1 atom g$^{-1}$ yr$^{-1}$ for each nuclide. All samples fall on or below the curve describing continuous exposure. Samples from the inner moraine set are consistent with continuous exposure and steady state erosion or fall below





the continuous exposure curve because of [10]Be inheritance (see below). Two samples from the two outer moraines display results consistent with continuous exposure, including one sample consistent with production-decay saturation (15-GH04). Most samples from the other moraines though are consistent with a history of complex exposure.

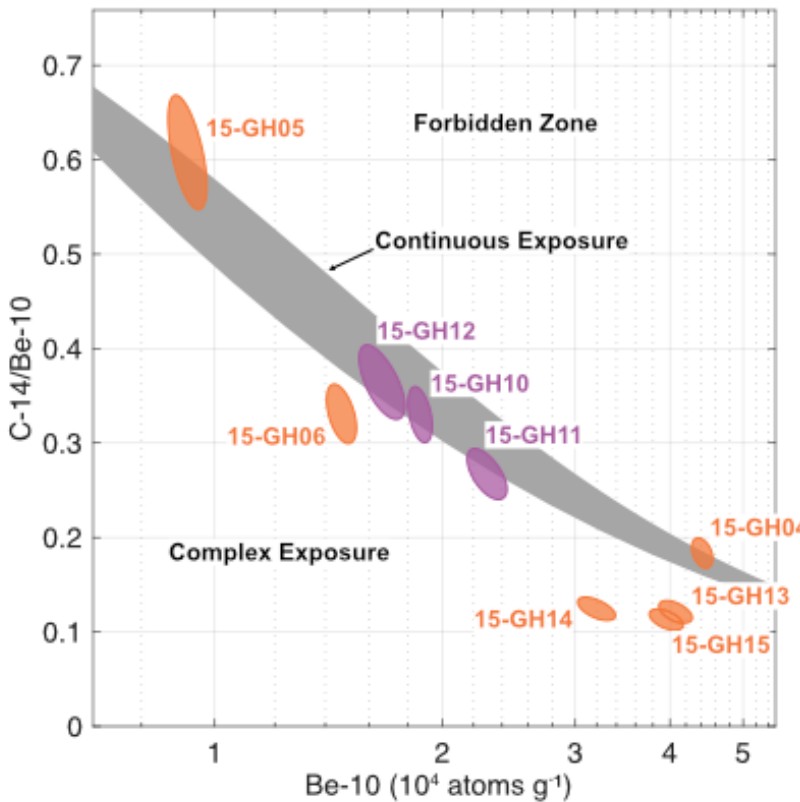

**Figure 6. Paired [14]C-[10]Be plot demonstrating that two outer moraine samples, GH05 and GH04, are consistent with a single-stage continuous exposure history with little or no surface erosion (upper bound of gray shaded region), the three inner moraines samples 165 are consistent with continuous exposure but also could be affected by a multi-stage exposure history with [10]Be inheritance, and finally four samples from the outer moraines are consistent with complex multi-stage exposure history. Note that [10]Be inheritance serves to lower the [14]C-[10]Be ratio and shift samples to the right. Outer (inner) moraine samples are shown in orange (purple).**

## 6. Discussion

The large degree of scatter observed in the [10]Be ages, as well as from any single moraine, argues for the presence of processes 170 operating external to climatic controls on glacier extent and moraine deposition. Below we will explore processes potentially leading to the observed [10]Be and [14]C concentrations/ages from a given moraine set. There are four possibilities affecting any given moraine dataset: 1) the apparent exposure ages are younger than the depositional ages as a result of erosion of the boulder surfaces during exposure of the sampled surfaces, 2) the apparent exposure ages are younger than the depositional ages as a



result of significant exhumation from within a moraine after deposition, 3) $^{10}$Be exposure ages are older than the depositional

age due to inheritance of nuclides from prior exposure, and 4) exposure ages and therefore moraine ages reflect real climatic

processes leading to moraine deposition.

Weathering and erosion of boulders via the removal of mass, either physically or chemically, lowers the measured

nuclide concentration from a boulder surface because of the advection of the future surface sample that accumulated

cosmogenic nuclides from a deeper depth in the rock to the surface. For an area such as Grey Hunter, where little gradient in

climate is observed across the study area and lithologic variations are also minor, and further limited by our selective sampling

of monzonite boulders, boulder surface erosion rates should be relatively uniform. For the purposes of surface exposure dating,

it is the rate of erosion that matters and is therefore presumed to be the same for both older and younger moraine samples.

Note that we are not implying that erosion rates necessarily are the same between glacial and interglacial times. All samples

are consistent with exposure subject to steady-state erosion (Figure 6); however, because of the wide range in apparent $^{10}$Be

exposure age for a given moraine rather than consistently too young, we discount boulder surface erosion as the cause for too

young exposure ages and do not discuss further.

First order observations indicate that the outer moraines have greater relative scatter than do the inner moraines. However, for

both the inner and outer moraines, the $^{10}$Be exposure age distributions are predominantly old-biased and thus indicate the

presence of inherited nuclides (Figure 4). A common approach to interpreting such age distributions is to use the youngest

sample from each moraine surface (e.g., Heyman et al., 2011). For the outer moraine sample sets (1, 3, 5, 8), this yields $^{10}$Be

ages of ca. 24, 23, 9, and 32 ka. Similarly, for the inner moraine sample sets (2, 4, 6, 7), we observe youngest ages of ca. 18,

15, 14, and 16 ka. The outer moraines using the youngest age approach are generally older than the inner moraines, but not in

all instances. Additionally, this approach is incompatible with the morphostratigraphic requirement of the inner moraine post-

dating the outer moraines within a single valley. Regardless, it is tempting to conclude that the outer moraines are a result of

early retreat from the LGM and the inner moraines a stillstand during recession from the LGM, like previous work (e.g.,

Margold et al., 2013a; Margold et al., 2013b; Stroeven et al., 2010; Stroeven et al., 2014; Menounos et al., 2017). Applying

the youngest age approach with the $^{10}$Be age is inconclusive at worst and unsatisfactory at best particularly given the

observation that sample set 5 yields ages from 9 to 44 ka.

We further investigated the processes operating at Grey Hunter and more robustly determined moraine ages by measuring in

situ $^{14}$C. In situ $^{14}$C should be less sensitive to inheritance given its short half-life. Thus, during periods of expanded ice extent

the combination of subglacial erosion and most importantly decay during ice cover reduces the number of nuclides present in

the sample at the time of deposition. Further, given the differences in depth-dependent production between $^{10}$Be and $^{14}$C, we

can investigate potential issues of exhumation of sampled boulders from moraine surfaces. This latter point is of potential

importance since many of the sampled outer moraine boulders were less prominent relative to their surroundings than the

sampled inner moraine boulders.

Before discussing paired $^{14}$C-$^{10}$Be results from outer moraine samples, we need to establish expected $^{14}$C concentrations and

$^{14}$C/$^{10}$Be concentration ratios. Our hypothesis with regards to moraine age means that all samples should return finite $^{14}$C ages.



Any resulting [14]C ages that are saturated (Figures 5 and 6) with respect to production decay systematics indicate that a boulder has resided at or near the surface for sufficient time (> 30 kyr) that production of [14]C is balanced by decay of [14]C. There is the potential for [14]C inheritance, implying that a sample was delivered to a moraine with sufficient [14]C to yield a saturated concentration in less than 30 kyr. We discuss this latter scenario first as we observe a single saturated sample from sample set 5.

We present two end-member model scenarios that could lead to inheritance in a moraine boulder sample, and we discuss model results in the context of our [14]C measurements (Figure 6). The first scenario involves the plucking of a pre-exposed surface from a valley bottom, and englacial incorporation of the resulting boulder and transport to its final moraine resting place (Figure 7A). The second scenario envisions delivery of a pre-exposed paraglacial boulder to a supraglacial setting prior to burial and englacial transport to a moraine (i.e., Scherler and Egholm, 2020; Orr et al., 2021; Ward and Anderson, 2010; Figure 7B). We use characteristic elevations for the valley bottom, cliff, and moraine settings and assume that in both scenarios, the [14]C concentration is at production-decay secular equilibria and thus represents the maximum [14]C concentrations that could be observed prior to delivery to the glacier surface. This means that as soon as the boulder is removed from its higher-elevation origin, there is excess [14]C that begins to decay and is no longer balanced by production as the new lower elevation has an attendant lower production rate.

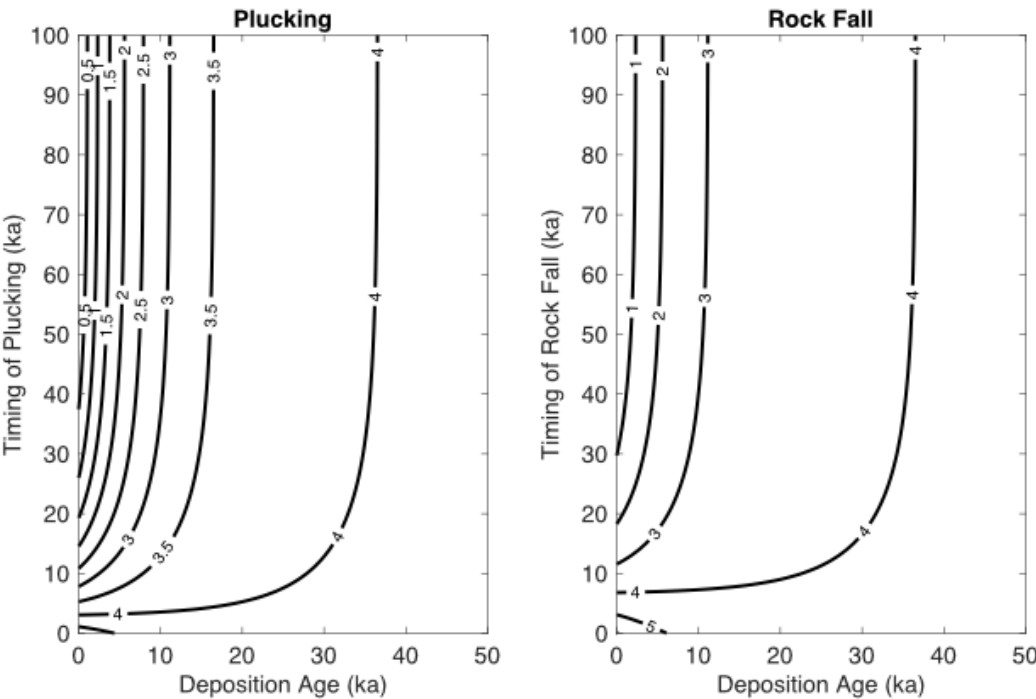

**Figure 7. A. Model of resulting [14]C concentrations for a scenario of plucking of boulder from 1000 m asl that was saturated with respect to [14]C prior to plucking. Contours show resulting [14]C concentration ($10^5$ atoms g$^{-1}$) for a range of plucking ages and moraine deposition ages. For reference, the saturation concentration for the mean older moraine elevation is 3.98 x $10^5$ [14]C atoms g$^{-1}$. B. As**



**in A, except that instead of the boulder being plucked by the advancing glacier, the boulder is delivered via rockfall from the headwall and transported supraglacially down valley before deposition on the moraine.**

Resulting theoretical [14]C concentrations for the above scenarios indicate that for nearly all deposition ages, the resulting [14]C

concentrations are effectively invariant with respect to the timing of plucking and entrainment or rockfall. This means that the observed [14]C concentrations will be almost entirely set by the age of moraine deposition, and only the youngest moraines (< 10 ka) would have any memory of pre-exposure. The above scenarios though are unrealistic for Grey Hunter given plausible ages for the inner and outer moraines, and thus we dismiss the possibility of inheritance due to pre-exposure affecting our [14]C measurements from either the inner or outer moraines, though not necessarily the paired [10]Be measurements. In contrast, we

conclude that the single saturated sample suggests relative antiquity (> 30 ka) for sample set 5, which is further supported by the ca. 44 ka [10]Be exposure age for sample 15-GH04, and a [14]C/[10]Be ratio indicative of continuous surface exposure.

So far, we have largely ruled out inheritance affecting [14]C measurements, thus allowing for some antiquity of the outer moraines. We also observe anomalously low apparent [14]C ages from the outer moraines (Figures 5 and 6), particularly relative to the inner moraines, along with [14]C-[10]Be ratios indicative of burial (Figure 6). This suggests that other processes are operating,

notably exhumation of boulder surfaces from eroding moraines, as boulders from the outer moraines lack prominence relative to their surroundings and appeared embedded, in contrast to the more prominent boulders of the inner moraine samples.

The exhumation of boulders from a degrading moraine will lead to apparent exposure ages that are less than the true depositional age of the moraine since the boulders accumulated nuclides at a lower rate than at the surface while buried. The degree of scatter for a set of exhuming boulders is expected to be large because of variable nuclide inheritance at the time of

deposition, but also because there is no a priori reason that the depth from which the boulder was exhumed need be consistent from sample to sample. We can use paired [14]C-[10]Be measurements to investigate the likelihood of exhumation processes occurring on the Grey Hunter moraines and leading to the observed scatter in apparent exposure ages and [14]C-[10]Be ratios indicative of burial. However, because of the strong likelihood of [10]Be inheritance resulting from exposure during multiple interglacial periods (Balco, 2011; Balco et al., 2014), we need to use [14]C-[10]Be ratios with caution. A more robust approach is

to exclusively look at [14]C concentrations since our model above effectively rules out the presence of [14]C inheritance. To explore this possibility, we model [14]C concentration resulting from exhumation within an eroding moraine followed by exposure on the moraine surface. Important to our approach here is that we account for [14]C production by muons, as the excitation function for production by negative muons (Heisinger et al., 2002a; Heisinger et al., 2002b) is large enough that significant subsurface production must be accounted for in all but the fastest of exhumation rates; to do so, we follow the simple approach outlined

in Goehring et al. (2013). We assumed a density 2.4 g cm$^{-3}$ for the eroding moraine and vary the duration of exhumation and the residence time on the surface, the sum of which represent the depositional age of the moraine and model resulting [14]C concentrations for a range of exhumation depths (Figure 8).

From our model, we can make some simple, yet powerful, inferences regarding the age of the outer moraines at Grey Hunter. For a boulder that has always resided at the surface of an outer moraine, the concentration is entirely set by duration of exposure

and follows the expected trajectory for a sample accumulating nuclides at the surface (Figure 6). For a given depositional age




(exhumation duration plus exposure duration), the deeper the exhumation depth, the faster the exhumation rate, and thus the less time spent at low production rates; the concentration is then more dependent on the duration of surface exposure.

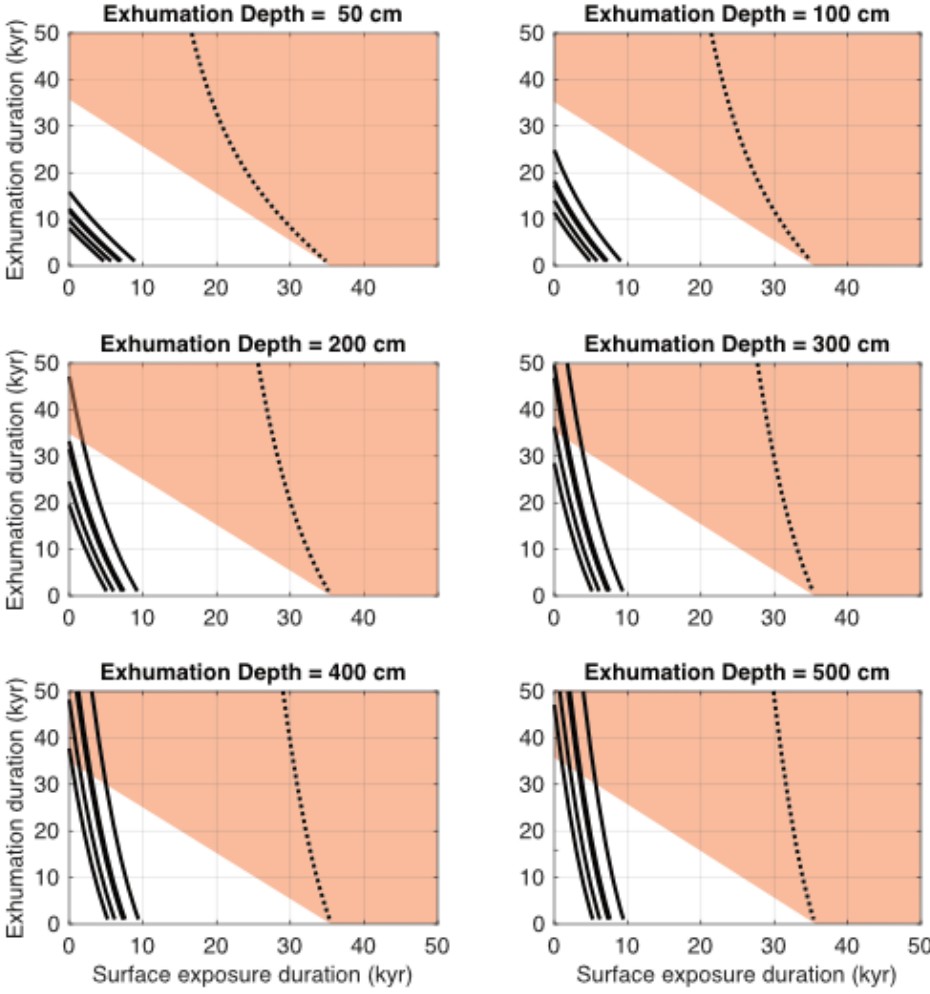

**Figure 8. Contours of measured ¹⁴C concentrations compatible with exhumation and exposure duration pairs for outer moraine**
**samples. Dashed black line represents theoretical and measured saturation concentration (15-GH04) and effectively sets the minimum age of the outer moraines given the unlikelihood of ¹⁴C inheritance. Orange shading represents the range of exhumation-exposure duration pairs compatible with saturated ¹⁴C concentration. The intersection of lower concentration contours with orange shading represents compatible exhumation depths and histories.**

Contours in Figure 8 outline exhumation-exposure duration pairs compatible with measured ¹⁴C concentrations. Additionally,
because ¹⁴C inheritance is unlikely (see above) we can use the single saturated sample to set minimum depositional ages for an outer moraine assuming it has resided at the moraine surface for its entire history. Doing so yields a field of compatible exhumation-exposure duration pairs assuming all samples have a minimum depositional age of approximately 35 ka (Figure





8). For the samples from the outer moraines with finite $^{14}$C ages we observe that one sample is potentially compatible with 200 cm of exhumation, while all other samples require at least 400 cm of exhumation over a minimum of 35 kyr. Additionally, the

saturated sample was deposited at least 35 ka, and older if exhumed from any depth. We can also infer that the young apparent $^{14}$C age samples must have $^{10}$Be inheritance from prior periods of exposure to yield the observed $^{14}$C-$^{10}$Be ratio, and that all samples reached the moraine surface during the Holocene.

To summarize, the scatter observed in both outer and inner moraine sets complicate interpretation and assessment of precise moraine ages. For the outer moraines, we are limited in our allowable inferences. Deposition pre-dates the LGM in its most

classical western-Cordillera sense and is 32 ± 8.2 ka based on the $^{10}$Be ages, regardless of the depth of exhumation, and is at least 35 ka based on $^{14}$C. These results mean that the outer moraines likely date to the classic definition of McConnell. Meanwhile, the tight clustering of $^{14}$C ages (11.2 ± 0.1 ka) from a single inner moraine, combined with the preponderance of young $^{10}$Be ages, suggests that the inner moraines post-date the LGM, and are correlative with major retreat events in the Canadian Cordillera at the same time (Menounos et al., 2017; Darvill et al., 2018; Lesnek et al., 2018). We are thus left with

the conundrum of the lack of apparent retreat from a set of moraines at the end of LGM in the canonical sense (i.e., 26-19 ka), but later retreat concordant with widespread retreat at the Pleistocene-Holocene boundary.

## 7. Climatic Implications

Our results suggest that the canonical LGM is absent, or at least not preserved in alpine glacier deposits at Grey Hunter. We cannot, however, rule out that there was an LGM advance smaller in magnitude and therefore overridden by the

advance associated with the inner moraines at the Pleistocene-Holocene transition. In contrast, the northern margin of the CIS appears to have classical LGM maxima ages and therefore there appears to be a different glacier response to past climate for Grey Hunter alpine glaciers. A possible explanation for the observed chronology is that of moisture starvation during the LGM at Grey Hunter. The presence of the LIS to the east-southeast induced cyclonic circulation over the general Yukon region, drawing down cold-dry air as katabatic winds off the LIS and from the north largely devoid of precipitation (Manabe and

Broccoli, 1985; Tulenko et al., 2020; Löfverström and Liakka, 2016). Thus, as climate deteriorated into the LGM, the expansion of the CIS and LIS hindered advance of the Grey Hunter glaciers beyond valley mouths. The expanded extent during MIS 3/4 in contrast is supported by elevated elemental carbon abundance during this time in Yukon permafrost, suggestive of greater productivity and generally wetter conditions in Yukon (Mahony, 2015). The same record also shows a marked increase in carbon content following the LGM and suggests that while temperatures likely remained colder than today during the latest

Pleistocene, precipitation relative to the LGM increased because of collapse of the CIS and LIS (Figure 9; Carlson et al., 2012; Liu et al., 2009), allowing for the expansion of glaciers at Grey Hunter.



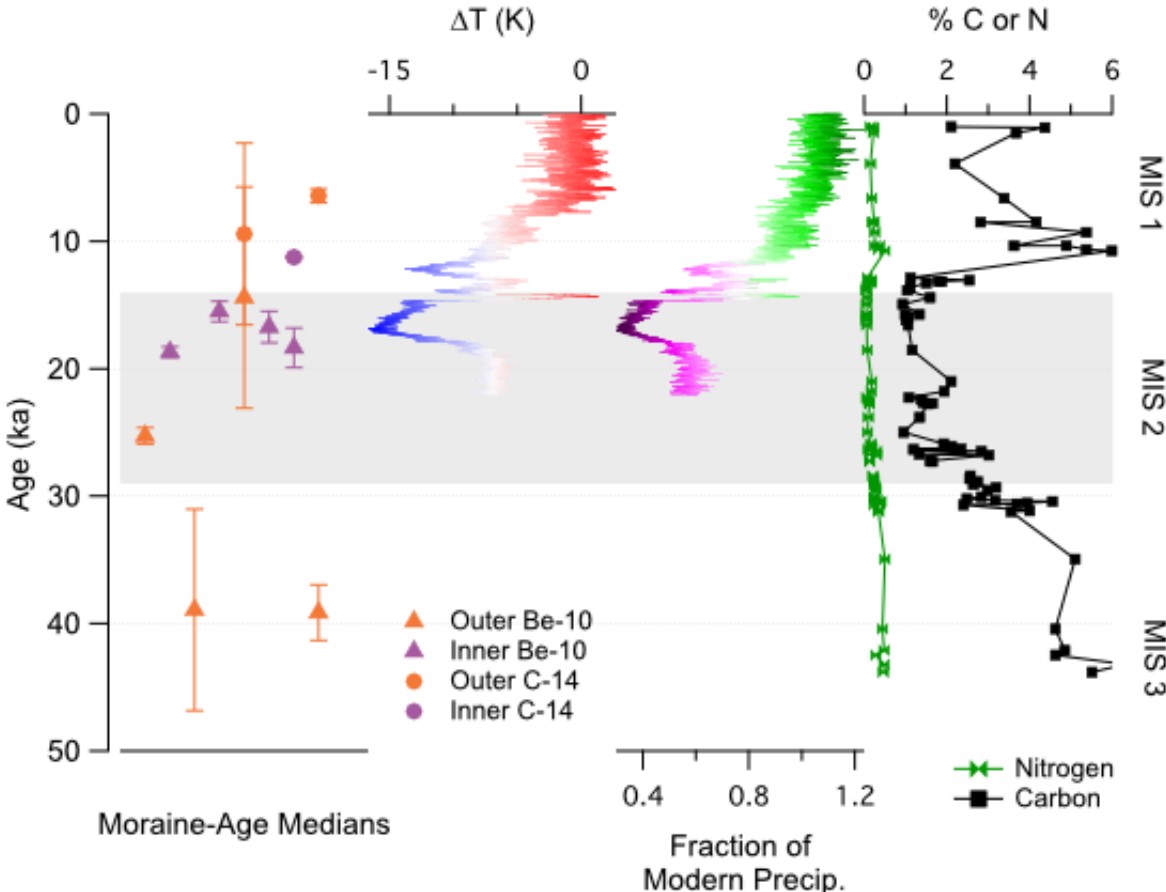

**Figure 9. A. Median ages of the inner and outer moraines for [10]Be and [14]C. B and C. TRACE 21k surface temperature difference and change in precipitation relative to present for the Grey Hunter region (Liu et al., 2009). D. Carbon and nitrogen content for the permafrost from the Yukon Territory (Mahony, 2015).**

Barring processes occurring post-deposition, most studies have identified inheritance as the largest contributor to scatter in surface exposure age populations from moraines (e.g., Heyman et al., 2011; Applegate et al., 2012; Applegate et al., 2010). Our results from Grey Hunter bear this out, at least in terms of the resulting [10]Be ages. The resulting [14]C concentrations (ages) however, cannot be explained by simple models of inheritance and instead require significant magnitudes of exhumation of boulders from within the moraine. Our exhumation model shows that the only parsimonious interpretation of results from the outer moraines is that they pre-date the LGM. The inner moraines on the other hand have clear [10]Be inheritance and young coherent [14]C ages, and date to the very latest Pleistocene/earliest Holocene. Our observations at Grey Hunter stand in contrast to the general observation of coherent surface exposure ages from other alpine glacier locations in British Columbia and the Yukon that were formerly covered by the CIS (Darvill et al., 2018; Menounos et al., 2017). A notable exception is the Glenlyon Range that largely avoided LGM CIS inundation and similarly has evidence for advance of alpine glaciers during MIS3 or earlier and at the Pleistocene-Holocene transition (Stroeven et al., 2014; Ward and Jr, 1992).



## 8. Conclusions

Our results are informative about not only the climate of the Yukon outside of the CIS limits, but also bear directly on the complications inherent in such a study. Our records from Grey Hunter massif suggest that conditions favorable for ice growth occurred during MIS 3/4 and at the Pleistocene-Holocene transition when temperatures were still reduced relative to the Holocene, but when the configuration of the CIS and LIS were such that enough precipitation reached Grey Hunter to induce positive mass balance. The cold-dry climate likely operating at Grey Hunter also means that glaciers were cold-based and resulted in the complicated exposure age distribution observed and that the use of [10]Be and [14]C yields information on processes leading to the observed exposure age distribution.

## Acknowledgements

We thank the staff of PRIME Lab and NOSAMS for their excellent AMS measurements for [10]Be and [14]C, respectively. Supported by grants from the Natural Sciences and Engineering Research Council of Canada (Discovery and Northern Research Supplement) and the Canada Research Chairs Program (Menounos) and Tulane University startup funds (Goehring). BMG thanks Chris Darvill and Greg Balco for fruitful discussion of ideas in this manuscript.

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
