# Peer review of "Reconciling the Apparent Absence of a Last Glacial Maximum Alpine Glacial Advance, Yukon Territory, Canada, through Cosmogenic Beryllium-10 and Carbon-14 Measurements"

_Geochronology, 2021_

## Author Response (AR1)

Response to reviewer comments and suggested changes

We thank Dr. Lamp for the constructive comments. They will no doubt improve the manuscript overall. Below, we have replied inline to review comments (general and line by line). Where comments are minor in nature (e.g., typos, etc.) we will make all suggested changes. Comments on figures are greatly appreciated, and we appreciate encouragement to include at least one table, as we will do in the final revised manuscript.

General Comments:

A table with (at least) the sample names, exposure ages, and 14C/10Be ratios should be included in the main text instead of the supplement. It would be helpful to see this information in a table near Figure 4, which would be easier to read and refer to than going through the text for individual exposure ages.

This is a good idea. We will add a summary table of sample names, inner or outer moraines, ages, and isotope ratios.

Can you comment on the likely source of the moraine boulders? If known, does this affect the applicability of your nuclide inheritance models?

Beyond a statement of up-valley we have no other constraints on the origin of the boulders sampled in this work. We are not sure that speculation would benefit or hinder the present manuscript.

Expand on the evidence supporting the division of moraines into two groups (inner vs outer). How do you know one or some of the moraines don't represent a different third advance?

This is a very valid a request, and we will expand on the logic behind our classification.

In order for readers to recreate the analyses, a table of the CRONUS-A data used to calibrate the 14C production rate is required (can go in the supplement).

All the CRONUS-A data are compiled in Goehring et al. (2019) as referenced. We argue that compiling again in the supplement is redundant when a reference to the full dataset is provided.

Near the end of the methods section where you describe how cosmogenic exposure ages were calculated, add that the ages are based on 0-erosion. How appropriate of an assumption is 0-erosion for these samples?

This is a very fair request and we will add such a statement. While certainly erosion is playing a part, the effect on 14C results will be negligible given production-decay systematics and the likely rate of boulder surface erosion operating. Effects on 10Be will be more pronounced, but not affect the overall conclusions of the manuscript. Thus we decline to explore or report further effects of erosion on resulting surface exposure ages, particularly because the argument of Holocene exposure means the boulder erosion rates would have operated on most of the samples for a very short amount of time, thereby not affecting the apparent exposure age significantly.

In the conclusion, local cold-based glaciation is mentioned; can you introduce that information into the text earlier on and expand on how that may or may not affect your measured exposure ages?

Yes, we can incorporate this more into the introduction so that it is set up better. The other reviewer made a similar request.

Figures

Figure 1: Add approx. dates for the 3 shaded glacial limits in the figure (either on the figure or in the caption). Add labels on the figure for some of the surrounding features/locations mentioned in section 1.2.1. Add north arrow + scale bar. In the caption: add a mention of the red box, e.g., "…delineating the Grey Hunter Massif (red box) as shown in Figure 2…". "Massif" is capitalized in the figure caption but is lower case throughout the text.

We appreciate the thorough comments on Figure 1, and along with the other reviewer will have a better figure as a result.

Figure 2: Indicate which moraines equate to which site numbers used in Figure 4 and throughout the rest of the text. Indicate which samples were analyzed for both nuclides. Needs north arrow. "Massif" capitalized again.

Thank you for the suggestion; we will incorporate all of them.

Figure 3: Add the names of the two samples in the images.

We will add sample names as request.

Figure 4: Again, add the site numbers used in this plot to Figure 2. Add the number of samples in each site group (n=X), perhaps above each box and whisker plot. In the caption, describe the components of the box and whisker plots (e.g., what do the triangles represent? What are those non-bolded thin vertical lines visible in some plots?, etc.)

Site numbers are represented by the bottom axis, but we can relabel to make clearer. We can certainly add the number of samples. We will add more details in the caption as requested.

Figure 5: Specify what the error bars represent in figure caption.

Error bars represent one-sigma uncertainties, but we will certainly add a statement to the caption.

Figure 6: Specify what the error ellipses represent in the figure caption. For the 3 inner moraine samples to be affected by 10Be inheritance, as mentioned in the caption, the amount of inheritance would have to minor for them to still overlap the simple exposure region, correct?

Error ellipses represent one-sigma uncertainties as well. We will add this to the caption. Yes, it is correct that the level of inheritance would need to be minor, as they would shift up and to the left if there was no 10Be inheritance.

Figure 7: Caption: Change "older moraine" to "outer moraine". What elevation did you assume for the headwall-derived saturated rock for scenario 2? See comment below for lines 216-217.

Thank you for correcting that. We will make sure all model assumptions are clearly stated in the main text.

Figure 8: Caption: add "(35 ka)" after "…minimum age of the outer moraines" in line 266. For clarity, in the next sentence ("Orange shading represents the range of exhumation-exposure duration pairs compatible with saturated 14C concentration"), specify that the range of compatible exhumation-exposure duration pairs are those that sum to >= 35 ka.

Thank you, we will add the anecdote of 35 ka and that the age/exhumation duration pair sums to 35 kyr.

Line by line comments:

We appreciate the detailed comments by Dr. Lamp. Where comments are small corrections or suggestions, we have made them as proposed. For more detailed comments, we reply in line below.

14: Remove "potential" (since you use potentially later in the statement)
15: Change "provide" to "provides"

71: Add "likely around" or "approximately" in front of -10.8C since it's an estimate 81: "dividing" instead of "breaking"?
107: If applicable: after mentioning the calibration data set add "….(sourced from http://calibration.ice-d.org/, accessed XX-XX-XXXX)"

Here, this is not applicable as calibration.ice-d.org does not yet contain 14C data. Instead, we refer readers to the 14C data in Goehring et al. (2019). For 10Be, we use the dataset detailed in Borchers et al. (2016), which is not compiled as a single dataset in ice-d.org.

115: LIA has not been defined yet; change to "Little Ice Age (LIA)"

159: You write: "Samples from the inner moraine set are consistent with continuous exposure and steady state erosion or fall below the continuous exposure curve because of 10Be inheritance." The way this is written makes it sound like at least 1 sample falls below the curve. However, error ellipses for all 3 samples fall within the continuous exposure region (and in the figure 6 caption it says "…the three inner moraines samples are consistent with continuous exposure"). Perhaps change to "…consistent with continuous exposure and steady state erosion with the possibility of minor amounts of 10Be inheritance."

We appreciate the suggested wording and will revise as such.

184-186: "…; however, because of the wide range in apparent 10Be exposure age for a given moraine rather than consistently too young, we discount boulder surface erosion as the cause for too young exposure ages and do not discuss further." The first half of this sentence reads as if you already concluded that the 10Be ages are not too young, while the second half reads as if you have already concluded that the 10Be exposure ages are too young. Replace with something along the lines of "…; however, due to the wide range in apparent 10Be exposure age for a given lithologically-homogenous moraine and no evidence for systematic underestimation of exposure ages, we discount rock surface erosion as a significant post-depositional process affecting apparent exposure ages."

We appreciate the suggested wording and will incorporate as such.

188: "…10Be exposure age distributions are predominantly old-biased…" based on what?

The 14C data? Specify.

This statement is based on the distribution as observed in the box-whisker plot. Given enough samples measured (not the case in our data), a PDF would show a right skew to the data, indicative of inheritance.

194: After "…within a single valley." specify the site with where the morphostratigraphic requirement isn't met. E.g., "…within a single valley (e.g., site X)."

216-217: The text says: "The second scenario envisions delivery of a pre-exposed paraglacial boulder to a supraglacial setting prior to burial and englacial transport to a moraine" whereas in the caption for Figure 7 it says the boulder is just transported supraglacially. Which is correct?

Both are correct, except that the caption neglects to mention the paraglacial exposure period. We will adjust the caption as such.

231-232: Add text in italics for clarity: "…deposition, and, *due to the short half-life of 14C, only the youngest moraines (<10 ka) would have any memory of pre-exposure.*"

Done.

233: Insert the "plausible age ranges" in parentheses for the inner and outer moraines.

We can certainly add those ages.

262-263: "For a given depositional age (exhumation duration plus exposure duration), the deeper the exhumation depth, the faster the exhumation rate, and thus the less time spent at low production rates; the concentration is then more dependent on the duration of surface exposure." I would rephrase this; even though the total time spent in the subsurface is less for the deeper samples, the deep samples spend some amount of time at lower production rates than shallower samples. Something along the lines of "For a given depositional age (exhumation duration plus surface exposure duration) the deeper the exhumation depth, the lower the effective or average nuclide production rate during exhumation largely due to the exponential decrease in spallation with depth. Therefore, for deeper samples a longer exposure duration is required at the surface (where consequence exhumation rates are faster."
We appreciate the suggestion and will rephrase accordingly.

273: Specify in parentheses the name of the "one sample" being referred to.

Done.

280: Is the 32 +/- 8.2 ka value based on the "grand group median" referenced earlier in the paper? State as such. In section 5.1, this value is quoted as 31.3 +/- 8 ka. Check and keep them consistent.

The value is indeed based on the grand group median and apologize for the rounding error. We will correct.

280: Move "regardless of the depth of sample exhumation" to between "western-Cordillera sense" and "and is 32 +/- ka." Otherwise it reads as if the date of moraine deposition is 32 ka regardless of the whether or not samples were exhumed.

Good catch, thank you.

281: Define the age range of the McConnell glaciation and add a reference.

We will add these ranges to the introductory and background material to avoid repetitive references.

282: Is 11.2 +/- 0.1 ka the group median again? Similarly to line 280, this value is written as 11.4 +/- 0.1 ka earlier in the text.

We will correct this error.

290-291: Add a reference for the statement: "In contrast, the northern margin of the CIS appears to have classical LGM maxima ages…"

References will be added.

294: Should "precipitation" be "moisture" instead?

Either word should work, but we will change to moisture.

298: Is a "the" missing in front of "Yukon"?

Yes, word added.

318: Move "not only" to be between "are" and "informative"

Done.

322: Change "positive mass balance" to "glacier expansion"

Agree in the context of the sentence, glacier expansion is a better phrase.

322-323: Expand on the statement: "The cold-dry climate likely operating at Grey Hunter also means that glaciers were cold-based and resulted in the complicated exposure age distribution observed…" Specify in what way cold-based glaciation at the site would have led to the exposure age observations. Also discuss this earlier in the text when first interpreting the data.

We will expand this idea earlier in the manuscript to allow for a return to this topic towards the end.

*We thank Dr. Tulenko for their insightful and useful comments. The comments and suggestions will no doubt make the manuscript better. Below, we have replied inline to review comments (general and line by line). Where comments are minor in nature (e.g., typos, etc.) we will make all suggested changes. Comments on figures are greatly appreciated, and we appreciate encouragement to include at least one table, as we will do in the final revised manuscript.*

General comments:

Readers would likely benefit from a deeper background/literature review/problem set up from the authors.

*RC1 had a similar comment and thus we will be expanding the introductory material a bit to accommodate topics revisited in the discussion.*

The authors use local nomenclature for ice advances (ie McConnel, Reid and pre-Reid) but there isn't information about what previous literature suggest the ages (relative or absolute) for those advances are. I suggest they define each local glacial advance that they reference and review relevant literature about the age of each advance.

*We will expand and define the state of knowledge for local glacial advances.*

It is not clear without looking through the figures what previous studies would suggest for the relative ages of the inner and outer moraines. Based on figure 1 it looks like the outer moraines were originally mapped as McConnell? Is that correct? I suggest making that clearer in the text and in Figure 2.

*We will clarify this in Figure 2, as Figure 2 will be significantly revised and improved.*

the authors mention that 10Be ages from other studies dating CIS deposits in the region are ambiguous but do not actually make any comparisons between results from this previous work and their new dataset. For example, how do the 10Be ages on the inner moraines compare to 10Be ages from previous work on CIS deposits? Are they comparable or not? I suggest discussing explicit ages from previous studies in the background or discussion section (or both).

*We disagree with this suggestion and decline going through a list of all of the previous dating efforts and rather refer the reader to past studies. Doing so will make the current manuscript too long. We feel we have already referred to the general ages that are inferred from past work.*

in the discussion, the authors mention studies from coastal AK and BC that show LGM advances, but do not say what ages those studies report for the LGM. I suggest the authors present that information in either the discussion or background section (or both) for comparison with their results.

*This comment is similar to the above comment. An exhaustive review of all previous work is beyond the scope of this research paper, and we do not want to write a review type paper.*

The audience would benefit from a little more clarity in the way they present their data through their figures and tables (see below comments on specific figures).

*Responses below with figure comments.*

I believe the discussion could be more complete by exploring other possible scenarios beyond their preferred interpretation.

What is the likelihood that the inner moraines were deposited near the end of the classically defined LGM? There appears to be some scatter in the 10Be ages, but enough clustering to suggest some moraines may have been deposited at the end of the LGM between 17 – 19 ka.

*We explored this option early on and largely moved beyond this because it would be entirely*

*coincidental and unlikely that exhumation processes, while certainly operating, would be such to exhume the boulders with measured $^{14}$C to the surface simultaneously. The most likely scenario is that of Lateglacial deposition.*

To this end, how representative are the 14C ages for all the inner moraines if they only come from one moraine?

*We cannot answer this question with any certainty given budgetary constraints and needed to select representative samples for analysis. We are not in a place to make additional measurements and elaborating would be pure speculation. Our analysis thus is grounded in the data we have in hand.*

Alternatively, if the inner moraines were all deposited sometime during the late glacial, what is the likelihood that the outer moraine was formed some time prior to the LGM and then re-occupied during the LGM? Is there any evidence from the new 10Be and 14C measurements to support or refute this hypothesis? For example, ages from moraine 1 show relatively low scatter and are within the timing of the LGM, is it possible that boulder ages from that moraine are representative of an LGM re- occupation?

*We argue that we explore this option in the current text, reoccupation is possible but unlikely and speculative. This is because such a LGM scenario and boulder deposition for some samples, but not others, requires $^{14}$C oversaturation (inheritance) and this scenario is unlikely given modeling (E.g., Figure 7). We will adjust the text to reflect this possibility more explicitly. But again, this becomes an exercise in selective data retention, while we sought in the current manuscript to seek explanation for the whole data set.*

If the authors believe either of these scenarios are unlikely, I would like to see them at least mentioned/addressed.

Figure comments:

Figure 1: Could the authors include a terrain/hillshade/DEM base map below the ice limits in this figure? Can the authors also include the mapped ice limits in the inset?

*We explored including a shaded relief map under the ice limits map and it initially did not work well. We will explore this option further. We decline to include ice limits on the inset as this is purely to serve as a location reference and would make the map too busy.*

Figure 2: It would appear based on figure 1 that the outer moraines are originally mapped as McConnel, is that correct? Could the authors overlay the ice limits from figure 1 onto this figure to make that less ambiguous?

*We will include the ice limits as poly lines in Figure 2.*

Is it also possible to somehow include 10Be and 14C ages onto map along with sample names so all the data is visible in one place? I recognize that might make the figure a bit busy, but if possible, could the authors do this?

*We debated this approach but found the figure to be far too busy. We appreciate the suggestion but decline to list ages on the map.*

Figure 3: In the text (beginning line 114), the authors first discuss inner moraines then outer moraines, and show Figure 3 in that order as well, but then outer moraine ages are presented first (line 130) followed by inner moraine ages (148). Could the authors fix this for general consistency?

*Yes, we will fix the order of presentation for consistency. Thank you for the suggestion.*

Related to figure 3. I was once asked by a co-author on this manuscript to share photos of all sampled boulders at least in a supplementary file, and I thought that was quite beneficial. If it is possible for the authors to do the same, I would recommend it.

*All photos of all boulder samples will be included in a supplementary Google Earth KMZ file. This should have been part of the submission package but apologize if it was omitted. The KMZ is included as an attachment here for reference.*

Figure 4: while I appreciate the box and whisker plot for each set of moraines to demonstrate the relatively high degree of scatter in some of the outer moraines, I feel this plot (or perhaps a second plot) would benefit from somehow displaying each individual age. Or perhaps, if considering all ages from inner moraines as one dataset and all ages from outer moraines as one dataset as the authors do, some histograms for inner and outer moraine ages may be appropriate. Could the authors find some way to show each individual age in a plot in the main text?

*We will explore options for displaying the individual ages and their corresponding uncertainties. One possibility will be a right hand panel showing probability density functions (PDF)for each moraine. We will also incorporate a similar PDF for the inner and outer moraines sample datasets.*

Also, I am unsure what each triangle for the moraine ages is supposed to represent. Please provide more detail in the figure description or on the plot.

*We will expand the figure caption to explain. Triangles represent samples considered outliers.*

Figure 8 and/or Line 272-274: it may not be necessary, but I would be interested to know total exhumation based on 14C and 10Be measurements if the (re-occupied) moraine were to be LGM in age (for example the mean age from moraine 1). Could the authors either do this and report values in the main text, or add an additional/supplementary figure?

*We decline to explore this option for the main reason that a LGM deposition scenario means that 15-GH04 is then overmatured with respect to $^{14}C$ production systematics and we showed this to be a nearly impossible scenario.*

Line by line comments:

*We appreciate the detailed comments by Dr. Tulenko. Where comments are small corrections or suggestions, we have made them as proposed. For more detailed comments, we reply in line below.*

Line 59: Fix this. Should that header be 2.1.1? If there aren't any other subsubsections here (although I think there could be), perhaps consider removing the subsubheader and place everything in just one subsection.

*Thank you, we will fix heading numbers for this and subsequent sections. We suspect we were done in by MS Word here.*

Line 75: Should this be section 3? If yes, then also fix the other subsections.

*See above.*

Line 107: can the authors justify the use of the default production rate from Borchers et al. (2016)? Do the authors argue that it is more representative of this site than other production rates (e.g. the Arctic Production rate from Young et al., 2013)?

*In this case, the use of the Arctic production rate, based on data entirely in the Atlantic basin, is less representative geographically than the global dataset that encompasses and wider geographic range. Further, the use of alternate production rates have no bearing on the interpretations herein. We thus decline to use an alternate production, rate nor justify the chosen production rate dataset, given that we declare the production rate dataset employed.*

Line 130: could the authors either report all individual ages here in the text or represent them in a table somewhere in the main text?

*All data are presented in the supplementary tables, and we decline to list every age within the text. We will be including tables of summary data but otherwise refer the reader to supplementary data for information on individual sample ages.*

Line 147: same comment as in line 130.

*Please refer to response to Line 130.*

Line 169: I might argue good coherence of 10Be ages in moraines 1 and 2, and moraines 4, 6, and 7 have at least two ages each that are somewhat coherent. I have certainly seen a lot worse in other places in Beringia. Can the authors do a little more justification here of not considering several 10Be ages?

*In this case, we are not examining the data on a moraine-by-moraine basis. Doing so then requires interpretation of individual ages, and subsequent rejection of the data. This argues against the main goals of the current manuscript, where we look for reason for discordia amongst a morphostratigraphic set of features.*

Line 274: See also comment on Figure 8. Is there any precedent for exhumation of ~4-5 meters since ~35 ka? If there is some literature on the topic, please cite.

*Given that the landscape after retreat was paraglacial, and thus likely ice-cored, rates of erosion are expected to be high. We will expand on this argument, including appropriate references. There are no specific studies focusing on this timescale, however that we are aware of.*

Line 280: can the authors justify the reason for averaging 10Be ages from the outer moraines? How likely is it that all outer moraines correspond to the same climatic event?

*Again, we are not attributing to any particular climate event. Rather, we are exploring the dataset in a conservative manner, guided by our modeling and ¹⁴C measurements. Our justification for averaging is guided by the similar ELA lowering associated with the outer moraines.*

Line 281: is there a typo here? Their preferred interpretation is that the older moraines are not McConnel in age, correct?

*That is correct, there is a typo, the word "not" is missing. Thank you.*

Line 290: could the authors include citations here, and perhaps explicitly report and discuss the evidence for classical LGM maxima ages along the CIS margin?

*Yes, we will add appropriate references. Thank you.*

Line 295: citations here are generally based on model results. If there is other terrestrial evidence the authors might lean on to suggest relatively dry conditions in the region during the LGM, please report.

*There are only modeling results referenced as most paleo-proxies are from regions that post-date deglaciation of the CIS/LIS, or were from deposits where preservation issues limit the reliable age range that data can be extracted from.*

Line 300: authors should report when the literature suggests the CIS-LIS saddle collapse occurred. How well do 14C ages from the inner moraine line up with the collapse? I feel the authors should spend a little more time discussing this idea.

*We are unsure why the saddle collapse (a specific region of the two ice sheets) is relevant here, as we*

*are generally referring to the broader CIS and LIS. Our references thus highlight observed changes in climate during CIS and LIS retreat.*